# From Immunotoxins to Suicide Toxin Delivery Approaches: Is There a Clinical Opportunity?

**DOI:** 10.3390/toxins14090579

**Published:** 2022-08-23

**Authors:** Matteo Ardini, Riccardo Vago, Maria Serena Fabbrini, Rodolfo Ippoliti

**Affiliations:** 1Department of Life, Health and Environmental Sciences, University of L’Aquila, 67100 L’Aquila, Italy; 2Faculty of Medicine and Surgery, Università Vita-Salute San Raffaele, 20132 Milano, Italy; 3Ministero dell’Istruzione, dell’Università e della Ricerca (MIUR), 20900 Monza, Italy

**Keywords:** suicide gene therapy (SGT), cancer, nanoparticles (NPs), extracellular vesicles, toxins, plant ribosome inactivating protein (RIP), modified RNAs

## Abstract

Suicide gene therapy is a relatively novel form of cancer therapy in which a gene coding for enzymes or protein toxins is delivered through targeting systems such as vesicles, nanoparticles, peptide or lipidic co-adjuvants. The use of toxin genes is particularly interesting since their catalytic activity can induce cell death, damaging in most cases the translation machinery (ribosomes or protein factors involved in protein synthesis) of quiescent or proliferating cells. Thus, toxin gene delivery appears to be a promising tool in fighting cancer. In this review we will give an overview, describing some of the bacterial and plant enzymes studied so far for their delivery and controlled expression in tumor models.

## 1. What Suicide Gene Therapy Is and How It Works

The curative approach against tumors has gained a wide interest in the last years as comprehensively reported [1,2]. A great impact on such scope is due to the so called “suicide gene” therapy (SGT), consisting of the selective delivery of genes coding for toxic proteins, into target cancer cells. This new and promising approach may overcome some of the issues related to the use of chemical agents (chemotherapy) such as aspecificity, high dosages with accompanying side effects and chemoresistance induction.

Cancer gene therapy may be thus approached using “suicide genes” within two possible alternatives: delivery of a toxin gene that is transduced directly into tumor cells inducing their death, or delivery of genes coding for enzymes modifying prodrugs, which in turn can release toxic metabolites (Gene-Directed Enzyme Prodrug Therapy, GDEPT).

Two crucial points need to be considered to result in a successful application of SGT. First, an ideal delivery system should allow the toxic enzyme or the prodrug activating enzyme to be expressed solely in cancer cells, with the limit of its expression level being sufficient to reach a minimal concentration of the toxin or the active enzyme to exert its toxic/enzymatic activity. Second, the targeted cancer cells that might express different levels of the suicide gene, so that in a complex tumor environment, a “bystander” effect might be desirable. The so-called bystander effect indicates transduction of the enzyme activity needed to induce the prodrug/toxicity in the neighboring cells, by transferring the cell death signals in the untransfected cells [3]. So, in this perspective, the toxin genes may be preferred for the first goal rather than prodrug activating genes that might be more suited for this second option.

Enzymes that have been studied as components of suicide gene constructs include viral and bacterial proteins such as viral thymidine kinase (TK), bacterial cytosine deaminase (CD), D-amino-acid oxidase (DAAO), bacterial carboxypeptidase G2 (CPG2), purine nucleotide phosphorylase (PNP), thymidine phosphorylase (TP), xanthine-guanine phosphoribosyl transferase (XGPRT), nitroreductase (NR), penicillin-G amidase (PGA), multiple-drug activation enzyme (MDAE), β-lactamase (β-L), horseradish peroxidase (HRP), β-galactosidase (β-Gal), and deoxyribonucleotide kinase (DRNK). To note, the specificity of substrate recognition and the absence of their counterpart in humans, allows us to avoid cross-enzymatic reactivity in substrate metabolization in human cells. However, as a major drawback, all the above-mentioned enzymes, being of non-human origin, may thus induce an immunological response. Human enzymatic activities may be also used in those cancer cells where the expression levels of these enzymes are significantly lower than in healthy tissues. Examples of these enzymes include β-glucuronidase (β-Glu), carboxypeptidase A (CPA), cytochrome P450 (CYP) and deoxycytidine kinase (dCK).

Substrates used for GDEPT as prodrugs have been recently described by Gholami and Sheikh et al. [1,4]. Among the above-described GDEPT approaches, the most widely studied and reported use CD and TK. Substrate prodrugs for these enzymes include 5-fluorocytosine, which is converted to 5-fluorouracil, and Ganciclovir (GCV), which is converted to Ganciclovir phosphate. Both these enzymatic products are then able to interfere with the DNA biosynthesis, blocking dTTP production (5-fluorouracile converted intracellularly to 5-FdUMP, 5-FdUTP or 5-FUTP) or accumulation of GCV triphosphate, an inhibitor of DNA polymerase. The catalytic activity of these enzymes requires the delivery into cancer cells of a prodrug, thus adding a second step to a successful suicide therapy approach. On the contrary, a direct approach by a single step through the delivery of genes coding for toxic proteins that do not need a prodrug would be highly desirable.

Various toxic agents have been studied in the last decades to be delivered into cancer cells as protein domains or DNA gene sequences; in these cases, the delivery is achieved by a chimeric construct formulation in which a specific carrier and the toxic enzyme or nucleic acid are the essential components [2,5,6,7,8,9,10]. With this regard, gene sequences coding for various potential therapeutic proteins have been proposed (i.e., DNAses, caspases, p53) as toxic components [11,12,13,14,15].

## 2. Genes Coding for Bacterial Toxins and Ribosome Inactivating Proteins as a Toxic Component of SGT

Genes coding for enzymes such as plant Ribosome Inactivating Proteins (RIPs; i.e., ricin, saporin, dianthin, gelonin, abrin, Pokeweed Antiviral Protein, PAP) or bacterial toxin domains (i.e., Pseudomonas exotoxin A, PEA or ETA and Diphtheria Toxin, DT) able to interfere irreversibly with protein synthesis, represent powerful tools in SGT (Figure 1).

The RIP enzymes are able to remove a specific adenine from a GAGA loop in 23S/25/28S rRNA (A4324 from rat 28S rRNA) thus irreversibly blocking translation (Figure 2). Otherwise, bacterial toxins such as ETA or DT catalyze transfer from NAD^+^ of ADP-ribose to elongation-factor EF2, thus inducing an irreversible protein synthesis arrest as a consequence of the inhibition of EF2 binding to the ribosome [16,17].

Some of these enzymes have been extensively studied in the past as toxic components of chimeric proteins called immunotoxins (ITXs) [16,17], made of a chemical or recombinant fusion between the toxin or toxic active domain and a targeting antibody or ligand binding domain. Note that, within the scope of this review, the term “immunotoxin” will be also used to indicate a broader type of chimeric toxins whose targeted domains are not exclusively made by antibodies. We recently placed emphasis on the advantages and disadvantages in the selection of ITX design and hosts for expression of chimeric protein toxins and use of recombinant immunotoxins [18]. The initial attempts to produce ITXs were associated to high costs of fully recombinant proteins and the related immunogenicity issues observed in treated patients as well as the risk of vascular-leak syndrome; furthermore, constraints due to low efficiency in cytosolic entry pathways of the toxin domains (where the target ribosomes are) were clearly observed. These issues led already in the nineties to the exploration of alternative gene delivery approaches, using selected toxic domains from potent inhibitors of protein synthesis, such as plant RIPs, or bacterial toxin. Active domains derived from *Corynebacterium diphtheriae* (DT-A) or *Pseudomonas aeruginosa* (ETA) moreover can act in a cell-cycle independent way, being thus able to kill both quiescent (i.e., tumor stem cells) and rapidly dividing cancer cells.

## 3. Bacterial Toxins

Amongst bacterial toxins inhibiting protein synthesis in therapeutic approaches, the DT active domain (Figure 1) is prominent. We wish to recall and acknowledge the three therapies approved by the Federal Drug Administration (FDA) for treating hematological malignancies based on bacterial toxin domains: Denileukin diftitox or ONTAK^®^, which was the first ligand-targeted toxin (IL2-DT) approved by the FDA (1999) and was recently dismissed, was followed by Tagraxofusp (IL3-DT, FDA approved in 2018). Both rely on DT-A toxic domain. Likewise, ETA/PEA derived from *Pseudomonas aeruginosa,* also called PE38, represents another prominent example of therapeutic immunotoxin. Recently approved by FDA in 2018 (brand name Lumoxiti) [19], is an ITX called Moxetumomab/Pasudotox (anti-CD22-ETA) for treating adults with relapsed or refractory hairy cell leukemia who have received at least two prior systemic therapies.

### 3.1. Diphtheria Toxin-Based Suicide Gene Therapy Approaches

More than two decades ago, Maxwell and his colleagues used for the first time a plasmid encoding the active domain DT-A for tumor ablation via viral gene therapy approaches, in vivo, using an SCID mouse model for B-cell lymphoma, pioneering the basis for tissue-regulated toxin suicide gene therapy [20].

In a similar perspective, more recently Peng et al. [21] have investigated the use of a chimeric modified enhancer/promoter of the human prostate-specific antigen (PSA) gene to regulate the expression of a DT-A-encoding DNA both in vitro in transfected human prostate cancer cells and in vivo xenografts derived from these tumor cells, as well as within tumors in TRAMP mice model. Likewise, the direct injection of the adenovirus-delivered DT-A gene into mouse prostates resulted in a dramatic reduction in the size of the gland [21].

As the studies on the use of toxin genes in SGT were approached, the issue of gene delivery and specific tissue expression was challenged early. With this regard, both viral and non-viral systems were screened. Cationic polymers like polyethylenimine (PEI) were later demonstrated to share several advantages compared to viral-derived vectors, including lower costs and easier production, higher versatility, lower immunogenicity/no systemic toxicity, ability to condensate larger DNA payloads into nanoparticles (NPs) and the possibility of inserting targeting peptides/aptamers or selected tumor-peptide specific domains. This may allow the Achille’s heel of selectively targeting the desired tumor cell to be overcome, which is a common obstacle to these cancer therapeutical approaches.

Another key strategy is promoters that can drive exogenous gene expression solely in the malignant tissue, to avoid leaky expression of the transfected suicide gene(s). Local delivery in the stromal environment or intratumoral gene delivery would seem the best options to avoid non-specific toxicities or, for certain tumor masses, after removal by surgery, for minimal residue therapy approaches. Targeted therapies may therefore be able to maximize anti-tumor efficacy while minimizing treatment-related toxicities, as often observed with the heavy side-effects due to chemotherapy first-line therapeutic approaches.

Among systematic approaches in DT-based SGT, a library of more than 500 degradable poly(β-amino esters) was generated and screened both to select for the best performing polymers in delivering toxin DNA locally, transfecting cells and accompanying cytotoxicity: C32 polymer displayed optimal characteristics for an effective local delivery (intratumor), lacking toxicity and avoiding transfection of the healthy muscle tissue. C32 delivered DNA intratumorally was 25-fold better than naked DNA and even 4-fold better than commercial jetPEI (polyethyleneimine) in expression levels. C32 was therefore investigated to transfect a DNA construct encoding the DT-A to a mouse xenograft derived from LNCaP human prostate tumor model. In addition, at the transcriptional level, suicide gene expression was regulated by a chimeric-modified enhancer/promoter of human PSA and the Flp recombinase. C32-mediated delivery of DT-A DNA not only efficiently suppressed tumor growth, but also resulted in a 40% tumor mass regression, revealing a powerful strategy to be adapted to other in vivo tumor models [22]. A similar approach was attempted using cationic biodegradable poly(β-amino ester) polymer as a vector for nanoparticulate delivery of the DNA encoding DT-A driven by two promoter sequences highly active in ovarian tumor cells, namely mesothelin and HE4. DT-A suicide gene was expressed specifically only within these tumor cells. Administration of DT-A nanoparticles directly to subcutaneous xenograft tumors or to the peritoneal cavity of mice bearing primary/metastatic ovarian tumors showed both a significant reduction in tumor mass and a prolonged lifespan of the treated mice, as compared to the controls [23].

Mesothelin is a target molecule in several tumors and is also specifically overexpressed in pancreatic cancer cell lines (CAPAN1 and Hs766T) but not in the surrounding healthy tissues. Through a novel biodegradable nanoparticulate system, mesothelin-expressing pancreatic cancer cells were targeted by DT-A expression. Resected pancreatic ductal adenocarcinoma specimens overexpressed mesothelin both at mRNA and protein levels. Luciferase gene reporter activity was measured in co-transfection experiments with the DT-A encoding DNA, indicating a great inhibition of protein translation (>95%) in mesothelin-expressing pancreatic cancer cell lines, when DT-A DNA was expressed under the mesothelin promoter. This strategy could potentially work in preclinical mouse pancreatic cancer models though it still needs to be confirmed in an in vivo modeling [24].

Tholey et al. [25] have both used the degradable C32 polymer or Lipofectamine 2000 to demonstrate that Mucin1 (MUC1), found overexpressed in pancreatic ductal adenocarcinoma (PDA) and being associated to tumor aggressiveness, could be targeted by transfection with a single dose of MUC1 promoter-driven DT-A. IFNγ pretreatment enhanced MUC1 expression in MUC1(-) cells and could then induce MUC1-DT-A sensitivity to this suicide therapy. Matched primary and metastatic tumor lesions from clinical specimens revealed similar MUC1 immunohistochemical labeling patterns, while a tissue microarray of human PDA biopsies showed an increased immunolabeling with a combination of both MUC1 and mesothelin (MSLN) antibodies, compared with either antibody alone.

This highlighted the need for additional cancer-specific promoters that target (i) a greater proportion of pancreatic cancers, and (ii) particularly the most lethal ones, possibly suggesting a combined approach with MSLN promoter-driven DT-A (as well as other specific promoter-driven constructs) in a multitargeted therapeutic approach that might be required to overcome tumor recurrence [25].

Another interesting approach involving DT as a therapeutic suicide agent comes from Huang et al. [23], which reported on the use of the DT-A toxin domain combined with NPs. The authors produced DT-A-encoding DNA conjugated with cationic poly(h-amino ester) NPs, which were injected into the peritoneal cavity of ovarian tumor-bearing mice resulting in a significant reduction of the tumor mass and increased animal life span with minimal nonspecific cytotoxicity, being much more effective than clinical doses of cisplatin and paclitaxel.

Finally, an interesting strategy is reported by Xu et al. [26] whose study shows the first light-inducible gene therapy approach using a DT-A cassette expression as a therapeutic suicide agent in killing malignant cancer cells. The authors developed a light-on gene-expression system to induce the expression of DT-A spatially and temporally upon illumination with a blue light of melanoma cells both in vitro and in vivo. The delivery system for the plasmid encoding light inducible DT-A are cationic liposomes combined with chitosan and further modified with a lipid, DOPE-PEG2000-cRGD. Similarly, the same group developed a light-switchable transgene NP-based delivery system in which NPs are made by a vitamin E succinate-grafted PEI-core and coated with an arginylglycylaspartic acid peptide (RGD)-modified PEGylated hyaluronic acid shell that mediates internalization via endocytosis. They demonstrated that B16-F10 melanoma cells were efficiently killed in vitro, as well as in the tumor-bearing C57BL/6 mice model, which showed a significant increase in the survival rate after this treatment [27].

Besides direct tumor injection, hydrodynamic gene delivery represents an alternative approach for achieving DT-based SGT in vivo. In this regard, Kamimura et al. [28] examined the antitumor effect of a DT-A-expressing plasmid in hepatocellular carcinoma cells (HCC) in vitro and in vivo by using the promoter of alpha-fetoprotein (AFP) for HCC cell-specific gene expression. By hydrodynamic gene delivery, that is a pressurized injection of the DNA-containing solution into the blood vessels, targeting the liver, they demonstrated that the overexpression of the AFP-regulated toxin results in diminished hepatocellular cell growth, through protein synthesis inhibition in an AFP-dependent manner.

Clinical trials have been started in Israel by several partners including private companies using DTA-H19 in phase 1/2a, dose-escalation, estimating safety, pharmacokinetics and preliminary efficacy study data in different tumors: intraperitoneal administration of BC-819 (H19-DTA) in subjects with recurrent ovarian/peritoneal cancer with cis-platinum resistance showed no major side effects attributable to the suicide agent. H19 is a long non-coding oncofetal ribo-regulator RNA expressed exclusively in certain tumors that has been intensively investigated in DT-A suicide gene approaches, by using targeted expression driven by the H19 promoter [29]. A remarkable study published in 2017 concluded that BC-819 is safe to use intraperitoneally in patients with ovarian, peritoneal and tubal cancer, however, since it was limited to a small number of patients, further studies should include larger cohorts, higher doses and longer periods of BC-819 treatment [30]. A previous phase 1/2a clinical trial using the same therapeutical approach to treat patients with superficial bladder neoplasm via intravesical delivery of DTA-H19/PEI complexes which was closed in 2007 has no results posted on the clinical gov site, while a second study, done in collaboration with the Maryland medical center, has results posted having enrolled 16 patients with unresectable pancreatic tumor for a Phase 1/2a DTA-H19 open labeled study that was concluded in 2010, in which seven of these patients could not, however, terminate this trial.

For a comprehensive excellent review of DT-A cassettes in suicide gene expression studies in preclinical animal models and clinical studies, please refer to Shafiee et al. [29].

### 3.2. Pseudomonas Exotoxin A/PE38-Based Suicide Gene Therapy Approaches

Concerning the use of Pseudomonas Exotoxin A (PEA), an example is reported by Schmidt et al. [31] that explored the GeneSwitch system, which consists of two plasmids: a regulatory plasmid, pSwitch, and the pGene/V5-His plasmid, in which they cloned the PEA active fragment (pGene/V5-His-ETA) to express exotoxin A in the hypopharyngeal carcinoma cell line, FADU. Stably transfected FADU cells were selected and the induction capacities of single pSwitch clones tested. Under the influence of experimental parameters such as transfection efficiency, the inductive capacities without antibiotic selection pressure and the inductive capacity after re-induction, constitutive expression levels were examined. In FADU cells the GeneSwitch-ETA combination worked precisely and effectively, suggesting this may be a promising approach for local gene therapy in head and neck cancers [31]. The same authors used an inducible expression cassette including the modified version of ETA lacking its cell binding domain, to induce ETA expression in head and neck cancer cells under the control of the progesterone antagonist mifepristone (RU486) and demonstrated that the target cells were effectively killed. However, the cells surviving the first treatment were then much less susceptible to induction, possibly due to integration of the DNA into an unfavorable region of genomic DNA, limiting further expression of the exogenous gene.

Khodarovich et al. [32] recently used the expression of Exotoxin A driven by the human telomerase promoter or by the ubiquitous CAG promoter (pTERT-ETA and pCAG-ETA) with a linear polyethylenimine transfection vector to target cancer cells. CAG is a widely used synthetic promoter, which derives part of its sequence from the chicken actin promoter/enhancer. Interestingly, these authors observed that selectivity of cancer cell killing by the pTERT-ETA was highly dependent upon the method of preparation of DNA-polyethylenimine complexes. Following changes of polyplexes preparation protocol, cell lines showing high activity of telomerase promoter were killed by transfection with pTERT-ETA plasmid. They showed that despite cells being transfected with pTERT-ETA and pCAG-ETA, plasmids do not exert any detectable bystander effect in vivo, and three intratumoral injections of plasmid-polyethylenimine complexes resulted in a substantial growth retardation of a poorly transfectable D2F2/E2 model tumor in mice.

A pCMV-ETA-EGFP DNA construct was transiently transfected with lipofectamine 2000 in HeLa cells where ETA-EGFP fusion protein to efficiently inhibit protein synthesis, leading to a caspase-3 dependent apoptosis-mediated cell death pathway. One interesting aspect of this study to assess protein translation inhibition in situ in Hela cells is the use of immunocytochemistry and a monoclonal antibody to the A3 antigen: this strategy allows the authors to follow the redistribution out from nucleoli of this component of the RNA polymerase I machinery, which translocate to the cytoplasm when HeLa cells are treated with known translation inhibitors [33].

Breast cancer SK-BR-3, MDA-MB-231 and MCF10A cell lines were transfected by using Polyamidoamine (PAMAM) dendrimers and constructs encoding a luciferase/PE38 under the control either of the CMV or/CXCR1 promoter with or without the insertion of a bFGF 5′UTR. Luc expression was evaluated using a dual-luciferase reporter assay, while PE38 expression was measured by real-time PCR and cytotoxicity determined by MTT assays, indicating a decrease in cell viability in the PE38 transfected breast cells, although an in vivo model will be needed to assess a safe and tumor specific restricted P38 expression [34]. In fact, this same group developed a functionalized PAMAM, with a multi-targeted nanosystem using anti-HER2 VHH (deriving from single chain variable library) coupled to CXCR1 promoter, PE38 toxin A gene and bFGF 5′UTR which selectively caused cytotoxicity in HER2-positive Breast Cancer Stem Cells. Overall, however, the data demonstrated that the efficacy of this targeted gene therapy was much lower in spheroid models of tumor, as compared to monolayer cell cultures. Being tumor spheroid models more like those observed in patients and/or preclinical animal models, the authors would predict a much lower antitumor efficacy for their nanosystem in vivo [35].

This points out the necessity of using mouse model of tumors to demonstrate the clinical effectiveness of these approaches, as it would be expected that in vitro expression of the toxic domains is always able to induce translation inhibition leading to apoptotic cell death. Thus, another relevant tool for preclinical studies is the in vivo modeling of human tumors. Two immunocompetent humanized mesothelin transgenic mouse lines were established as tolerant hosts for C57Bl/6-syngeneic cell lines expressing a human isoform of mesothelin. Thyroid peroxidase (TPO) mice have thyroid-restricted human mesothelin expression. Mesothelin (Msl) mice expressed human mesothelin typically in the serosal membrane and were used to assess on-target, off-tumor toxicity of human mesothelin-targeted therapeutics. Both the transgenic strains shed human mesothelin into the serum, similarly as in human mesotheliomas and in patients affected by ovarian cancer: serum human mesothelin can be used as a blood-based surrogate of tumor burden. In these models, the authors examined the on-target toxicity and antitumor activity of human mesothelin-targeted recombinant ITXs. Most importantly, they have defined two high-fidelity, immunocompetent murine models that mimic human cancers allowing for rigorous preclinical evaluation of human mesothelin-targeted cancer therapeutics [36].

Investigating the combination of a tumor-specific promoter to drive a specific toxin domain expression is still an opportunity to examine, as reported herein. An in vitro approach explored the SERPINB3 gene (highly active in oral squamous cell carcinoma): SERPINB3 promoter-mediated PE38KDEL expression vector transfected with PEI (C202H505N101) was tested in different cell lines TCA8113 (tongue squamous cell carcinoma), MG63 (osteosarcoma), Eca-109 (esophageal cancer), HeLa (endocervical adenocarcinoma) and MCF-7 (breast cancer). SERPINB3 RNA and protein were expressed at low levels in MG63 and L02 cells but highly expressed in TCA8113 cells, as expected. Unexpectedly, although the SERPINB3 protein was barely detected in HeLa cells, the corresponding mRNA was found expressed at a high level. SERPINB3 promoter activity was detected by luciferase assay and the suicide vector pSERPINB3-PE38KDEL was tested in the SERPINB3-positive TCA8113 cell line, and in the controls. Transfection of the pSERPINB3-PE38KDEL plasmid effectively inhibited cell proliferation and induced cell apoptosis, with no significant damage to MG63 and L02; however, a transwell invasion assay showed that significantly fewer TCA8113 cells than MG63 cells passed through the gel matrix following pSERPINB3-PE38KDEL transfection [37].

### 3.3. Other Bacterial Toxins to Be Mentioned as Related to SGT

Beyond the group of toxins irreversibly inhibiting protein synthesis, Walther et al. [38] used a plasmid carrying the gene for *Clostridium perfringens* enterotoxin (CPE) to target cancer cells overexpressing claudin-3/-4. Using a commercial transfection agent (Fugene™) the plasmid sequence carrying a codon-optimized CPE coding sequence was efficiently transcribed and translated into the cells. The expressed toxin was released in the extracellular space, inducing cell lysis in cancer cells as CPE can bind claudins to form a pore in extracellular membranes. Since many tumors overexpress the claudin receptor, this targeted cancer therapy has deserved further investigation in animal tumor models.

The same CPE toxin selectivity towards claudin-3/-4 was further studied by the authors of [39], who have characterized poly(lactic-co-glycolic-acid) (PLGA) NPs modified to bear a COOH-terminal domain binding CPE for the delivery of DT-A to chemotherapy-resistant ovarian cancer cells, under the transcriptional control of an ovarian specific p16 promoter, which is highly differentially expressed in ovarian cancer cells. The control plasmid DNA was GFP under CMV promoter to obtain GFP NPs or with the targeting of CPE (CMV GFP c-CPE-NPs). Both these particle formulations showed an initial slow release of the DNA followed by a burst of between 12–72 h of incubation in the culture medium at 37 °C. OSPC-ARK-1-derived xenografts were treated with vehicle, or NPs c-CPE-NP encapsulating the mock or the p16 DTA c-CPE for 30 days and monitored for overall survival (OS) for a total of 45 days after the first treatment. The suicide gene containing NPs, p16 DT-A c-CPE-NPs, significantly improved the survival of tumor bearing mice when compared to control vehicle injected mice (*p* = 0.007) or mice injected with mock control c-CPE-NPs.

Finally, the use of the CPE gene to transfect human colon cancer cells overexpressing claudin-3 and claudin-4 [40] was also reported. The binding of CPE to these proteins triggered the formation of a multi-protein membrane pore complex inducing loss of osmotic equilibrium and finally cell lysis. The transfection greatly reduced the growth rate of colon cancer in in vivo studies. Optimized DNA sequence coding for CPE was efficiently transferred to cancer cells where massive CPE expression led to its release in the extracellular environment, also allowing an efficient bystander effect of CPE on neighboring cancer cells.

Another remarkable example of SGT is reported for Streptolysin O (SLO), a hemolytic exotoxin that belongs to a large family of cholesterol-dependent cytolysins having pore-forming activity. A conventional plasmid expression vector carrying the SLO gene in combination with a liposome-mediated transfection vector [41] has been developed, causing necrosis of the targeted cells by creating large pores at the cell membrane. The first example reported on HEK293T (human embryonic kidney fibroblast) cells showed that transfection with the SLO-liposome system led to cell membrane permeabilization and disintegration, causing the cells to die. The same group further developed an adenoviral expression vector, for a high-efficiency transfer system to express the SLO gene. They showed that this vector significantly reduced the viability of several human cancer cell lines, such as cervical and lung carcinomas, breast and prostate cancer cells, as well as tumor engrafted cells in CA33 xenograft-bearing mice.

Bacterial toxin-antitoxin (TA) systems may represent a reservoir of toxins that might be used in SGT. TA systems are expression modules where a toxin operon is found co-expressed with its cognate antitoxin module. The ldrB gene belongs to this group of toxins and has been recently evaluated for its ability to kill cancer cells through SGT, as shown by Jiménez-Martínez et al. [42]. In this study, the authors developed an HCT-116 colorectal carcinoma and MCF-7 breast cancer cell lines treatment by transfecting the Tetracycline inducible expression system Tet-On 3G carrying the *E. coli* ldrB gene. The results demonstrated the inhibition of cell proliferation by the induction of apoptosis in vitro and in vivo within xenografts in mice for both tumoral models when expression was induced with the Tetracycline analogue, doxycycline.

A major human pathogen, *Staphylococcus aureus* producing the Enterotoxin H (she), can cause diseases by inducing apoptosis through programmed cell death, thus representing another promising toxin to be considered for SGT-based therapy. An interesting example is reported by Safarpour-Dehkordi et al. [43] who showed for the first time the expression trend of long non-coding RNAs (lncRNAs) during tumor progression in renal cell carcinoma and upon in vitro transfection of ACHN (metastatic renal carcinoma) and PC3 cancer cell lines. They used a eukaryotic expression plasmid pcDNA, including the SEH-encoding gene and transfection with lipofectamine 2000. Their results demonstrated how the transfection caused an up-regulation of some apoptosis-related lncRNA and therefore an increased cell death.

To be especially noted, injections of vectors including synthetic modified mRNA coding for toxins might become useful tools instead of whole DNA plasmids to stimulate target cell death in tumors. An original example is reported by Hirschberget et al. [44] that have developed an SGT approach based on chemically modified RNAs (cmRNAs). In this study the cytotoxic potential of cmRNA coding for DT-A, the subtilase cytotoxin from *Escherichia coli* (STEC) or the plant-derived RIP Abrin-A isolated from *Abrus precatorius* L. have been all tested in vitro and in vivo with the cmRNAs of Abrin-A being the most active out of the three tested via several intratumoral injections on KB models (human epithelial carcinoma cells) and Huh7 (hepatocyte-derived carcinoma cell). The authors concluded that this RNA-based SGT technology is a valid alternative to explore, especially since it is showing safety-relevant benefits, as compared to ITXs that may cause hepatotoxicity, or to DNA-based therapeutics which bear the risk of insertional mutagenesis. This is an open question which may deserve a deeper investigation, especially based on the great success obtained by the scientific community by using the modified mRNAs recently investigated by Katalin KariKo’s group in SARS-CoV-2 vaccination approaches [45].

A special mention is due to the so-called double-enhanced SGT. The first example has been shown by Boulaiz et al. [46] using a couple of two cell killing genes, namely gef from *Escherichia coli* and apoptin from chicken anemia virus, to inhibit the cell growth in DLD-1 colon carcinoma cells [46]. The authors produced DLD-1 cells co-transfected with the regulatory vector pRevTet-On and retroviral vectors containing gef, apoptin or both genes and evaluated the growth trend upon expression induction with tetracycline or doxycycline. The results showed that cells co-transfected with both gef and apoptin increased cell necrosis by enhancing the cell cytotoxicity apoptosis, likely via mitochondrial pathway, which may be deficient in colon cancer.

### 3.4. Novel Approaches

As many SGT approaches exploit DNA constructs to be delivered, it is worth noting that delivery of toxin gene mRNA could represent a smart alternative. In this perspective there is huge amount of literature and as an example it has been reported in a phase 1 trial on the safety and efficacy of mRNA-1944, a lipid nanoparticle-encapsulated messenger RNA encoding the heavy and light chains of a specific monoclonal neutralizing antibody, CHKV-24 (NCT03829384) against Chikungunya virus (CHIKV) infection which causes severe acute disease and has no therapy nor prevention treatments. The adult healthy participants received intravenous single doses of mRNA-1944 or placebo at 0.1, 0.3 and 0.6 mg/kg, or two weekly doses at 0.3 mg/kg. At 12, 24 and 48 h after single infusions, dose-dependent levels of CHKV-24 IgG with neutralizing activity were observed at titers predicted to be therapeutically relevant concentrations (≥1 µg/mL) across doses that persisted for over 4 months whereas minor adverse effects (infusion-related) were mild to moderate in severity and did not worsen following a second mRNA-1944 infusion [47].

Pseudouridine (Ψ) is the most prevalent modified nucleoside that is found in RNAs, whose main function seems to be stabilizing the crucial secondary structure of Ψ at specific locations in tRNA and ribosomal RNA (rRNA). In vitro transcribed mRNA containing those modified nucleosides was shown to be much less stimulatory to several host defense RNA sensors, including protein kinase R (PKR), toll-like receptor (TLR)3, TLR7, TLR8 and retinoic acid-inducible gene I (RIG-I). Katalin Karikò reported that production of in vitro transcribed mRNA where uridine is replaced by pseudouridine (Ψ-mRNA) drove protein expression higher than an unmodified in vitro transcribed mRNA, demonstrating that this enhanced translation is in part dependent from lesser activation of PKR by Ψ-mRNA of the immune system [48].

This technology could represent a valid and attractive alternative to DNA vectors to be explored in animal tumor models, since the employment of toxin mRNAs may have several advantages as compared, for instance, to plasmid-driven expression of toxin genes: the reduced size and different structure of this nucleic acid (that does not need to cross the nuclear envelope) is translated immediately when it reaches the cytosolic compartment, among other advantages. In addition, by using mRNA the risk of genomic integration is avoided (which might potentially lead to mutagenesis events), thus inducing a lower toxicity associated with the intracellular expression of toxic domains that do not need to be transcribed. A particularly interesting approach is reported by Guimaraes et al. [49] in designing a library of engineered ionizable lipid NPs (LNPs), which have been pooled and directly screened in vivo for optimal delivery in multiple organs/tissues of traceable barcoded mRNAs in their 3′ untranslated region (UTR), which allows for their direct quantification using deep sequencing. These b-mRNA are similar in structure and function to regular mRNAs but they contain barcodes and a unique molecular identifier (UMI) to avoid any misidentification. Deep sequencing results were validated via LNP delivery of a reporter luciferase mRNA, showing that the platform can effectively identify lead LNP formulations for mRNA delivery in vivo to organs such as the liver and spleen. In addition, when they compared delivery to the liver or spleen of b-mRNA to barcoded DNAs, they demonstrated that selectivity of LNPs might also be, surprisingly, dependent on the nature of the nucleic acids delivered [49].

## 4. RIPs Plant Genes as Toxic Weapons

As bacterial toxins are well known as tools for SGT approaches, plant ribosome inactivating proteins (RIPs) are becoming a valid alternative, even though no therapeutics based on plant RIPs were approved, despite some promising clinical trials having been conducted in the past (please refer to D. and S. Flavell’s review article in this Special Issue). Among SGT approaches, Bai et al. [50] have described the use of a RIP-coding sequence (Gelonin from *Gelonium multiflorum* L.) to prepare a gelonin-based nanocomplex in which the toxin gene is placed in an expression plasmid (pVAX1, under the control of the strong CMV promoter) which was complexed to a soluble Heparin-PEI nanogel for intracellular delivery. Optimal concentration was determined at 2 µg of gelonin DNA/10 µg HPEI. They demonstrated the cytotoxic effect of this construct to SKOV3 ovarian cancer cells both in vitro and in vivo, where they found a significant reduction of volume of tumors without signs of toxicity to healthy surrounding tissues, when using a total of 5 µg DNA/25 µg HPEI. Apoptotic cell death of tumor cells from sections after this treatment was observed, suggesting a potential anticancer therapeutic for local application.

Min et al. [10] reported the construction of mammalian expression plasmids carrying the RIP genes coding either for Saporin (from *Saponaria officinalis* L.) or gelonin, again under the control of the strong CMV promoter. These two plasmids have been used to efficiently transfect several cancer cell lines using PEI as a lipid carrier, demonstrating that toxin genes were able to kill almost all the cells after 48 h of protein synthesis inhibition. Optimization of the transfection conditions with PEI polyplexes led the authors to a plasmid concentration of 4 µg/mL with a plasmid/PEI ratio 1/10, to control cytotoxicity. Sama et al. [51] showed that peptide-based nanoplexes can be used for cancer therapy. The nanoplexes carry a positively charged peptide bearing a further receptor-directed sequence and “sapofectosid”, a triterpenoid saponin extracted from the plant *Saponaria officinalis* L., which enhances the endosomal delivery of both RNA and DNA such as the one coding for saporin. The authors demonstrated that these constructs have a marked anti-tumoral effect against neuroblastoma cells in vitro in co-transfection experiments of Neuro-2A-Luc-cells (murine neuroblastoma cells), with 2.5 µg/mL sapofectosid, with successful results also in vivo on Neuro-2A-Luc bearing NMRI nu/nu–mice.

A step forward integrated in this technology is represented by the so-called “nanoplasmids”, made of minicircle DNA constructs. This technology is being considered highly innovative in terms of pharmaceutical applications though, however, it still has a major drawback being expensive in terms of production costs. A very recent example of RIP-containing nanoplasmid for SGT is reported by Mitdank et al. [52] who investigated the use of two RIP toxin genes (i.e., Saporin from *Saponaria officinalis* L. and Gypsophilin-S from the *Gypsophila elegans* L.) to construct DNA nanoplasmids. The authors used formulations including size-reduced plasmids (nanoplasmids) that were characterized for their ability to be transfected into murine neuroblastoma cells using a cell targeting peptide (K16 lysine rich) and a lipidic permeabilization agent and showed a partial but consistent decrease of tumor growth in vivo. An interesting aspect of this study is the apparent absence of toxicity during this treatment, at least at the doses used in these tests, with both suicide nanoplasmid vectors carrying either Saporin or Gypsophilin-S genes.

Recently, the saporin gene was delivered for the first time intratumorally in the BL6 melanoma mice model which was complexed with either DOTAP or PEI [53,54] and obtained a great reduction in tumor masses in this aggressive mouse model. In addition, a more recent study [55] described the successful targeted expression of the gene encoding Saporin in human glioblastoma cells, by using a modified DNA plasmid carrying the double stranded sequence of the aptamer AS1411, specifically targeting glioma cell-surface localized nucleolin. The construct (dsDNA) was able to specifically target U87 model glioblastoma cells, not affecting the viability of non-tumor cells (mouse 3T3 cells). This is the first example of gene targeting by using an aptamer sequence directly embedded into a dsDNA construct, being improved (lower doses needed) thanks to the use of PEI as lipidic carrier in the DNA polyplexes. The Saporin gene was transcribed upon plasmid absorption, while the toxin mediated cell death activity was determined following an unusual mechanism, called methuosis.

Another interesting example is reported by Piña et al. [56], which showed an innovative RIP-based nanosized complex including elastin-like recombinamers (ELRs), MUC1-specific aptamers and a DNA plasmid encoding for the plant RIP PAP-S. This nanosized system can be selectively internalized via macropinocytosis within MCF-7 breast cancer cells while protecting normal cells and causing death of the malignant cancer.

According to the most prominent aforementioned examples, Table 1 aims to provide an overview listing the main prodrug-activating enzymes, bacterial and plant toxins used in SGT discussed throughout this review.

## 5. Nanoparticle Vector Systems for Targeted Delivery of Toxic Genes

Vectors were usually derived from viral particles for their well-known ability to infect cells and release their genetic content, i.e., the exogenous gene of choice to be delivered in this case, to the host cell, as we mentioned above. A number of viral vectors for SGT have been modified and used in both experimental settings and clinical trials [57].

Novel models for delivering exogenous genes have been developed in the past few years, by taking advantage of non-viable particles. Next-generation delivery systems rely mostly on nanotechnology. In this regard, nanosized objects with supramolecular architecture are becoming pivotal as effective therapeutic tools: they combine high efficiency of targeting by a combination of inorganic or organic NPs together with the power of the suicide genes to be delivered. The use of NPs in biomedicine is a very active field of applied medical research during the last 15 years. NPs have been shown to be very efficient in drug delivery, intracellular tracking and imaging without causing cell damage or tissue toxicity. A useful remarkable feature of NPs is their high surface area to volume ratio that allows them to interact with living matter as they are tools for extended chemical and biochemical surface functionalization, leading to a selective interaction with the molecular targets, e.g., the tumor tissue. Almost every kind of biomolecule has been linked to the surface of an NP, be it organic or inorganic, including DNAs, RNAs, proteins, enzymes and antibodies, aptamers, oligopeptides and oligosaccharides, the latter of which may serve also to reduce toxicity and increase NP stability in biological fluids, as reported by Lunova et al. [58] and Yang et al. [41].

Nowadays, NPs are considered valuable tools in both diagnostics and therapeutics with an increasing number of examples in biomedical applications as well as in oncology now being reported. NPs, indeed, can be easily produced or even selected from commercial suppliers ready-to-use in terms of size, electrical surface charge and, most importantly, surface functionalization, allowing for the bioconjugation with therapeutic moieties such as genes or proteins. An example of NPs-based original suicide therapy was presented by Paris et al. [59] who showed how Decidua-derived Mesenchymal Stem Cells (DMSCs) could be used as a Trojan-horse for cell mediated cancer therapy approach in vitro against NMU cancer cells. They transfected DMSC cells using non-viral agents, such as polycation-coated Ultrasound-Responsive NPs (UR-NPs). The most successful NPs formulation was then employed to induce the expression of two suicide genes: cytosine deaminase and uracil phosphoribosyl transferase, which allow the cells to convert a non-toxic prodrug (5-fluorocytosine) into a toxic drug (5-Fluorouridine monophosphate) that was evaluated in the NMU cells co-cultured with the transfected vehicle cells.

Another example is reported by Davis et al. [60], who showed NPs-mediated SGT targeting hypoxia-specific expression of therapeutic cargoes such as the herpes simplex virus thymidine kinase (HSV-TK) suicide gene or the gene for CRISPRCas9 nuclease. Hypoxia is a characteristic feature of solid tumors contributing to tumor aggressiveness being often associated with resistance to cancer therapy. In this case, five hypoxia-responsive element (HRE) sequences were inserted in the promoter of target genes encapsulated and delivered by lipid nanoparticles (LNPs) to achieve specific killing of tumor cells in hypoxic conditions. The authors showed high transgene expression in cells in a hypoxic environment, similar to levels achieved using the cytomegalovirus (CMV promoter), while showing no significant effects on cell viability in normoxia. Besides their effectiveness, these examples and many others in the literature report on how the suicide gene therapy using non-viral vector systems could be very promising, especially concerning safety reasons which must be considered when using viral vectors.

Farokhzad et al. [61] explored a chemotherapy approach using Docetaxel (Dtxl)-encapsulated NPs combined with biocompatible and biodegradable poly(D,L-lactic-co-glycolic acid)-block-poly(ethylene glycol) (PLGA-b-PEG) copolymer and a functionalized fluoropyrimidine RNA aptamer (Apt) to recognize the prostate-specific membrane antigen (PSMA) of LNCaP prostate epithelial cells. After a single intratumoral injection of Dtxl-NP-Apt bioconjugates, complete tumor reduction is observed in five out of seven LNCaP xenograft nude mice with 100% survival in the 109-day study, whereas only two out of seven mice in the Dtxl-NP group showed complete tumor reduction with 109-day survivability of only 57%.

Regarding nanostructured complexes for therapeutic uses, the aforementioned example reported by Piña et al. [56] is worth noting. In this study, the authors prepared a nanosized protein-based complex made by lysine-enriched elastin-like recombinamers (ELRs). These complexes showed a particle size of 140 nm and a positive zeta potential of approximately 40 mV able to interact with MUC1-specific aptamers and a DNA plasmid encoding for the plant RIP PAP-S, resulting in increase of transfection specificity for MCF-7 breast cancer cells. These hybrid polyplexes have been shown to penetrate the cells mainly by macropinocytosis and causing death of the malignant cancer while being inactive against healthy cells.

Figure 3 (panels A and B) shows the morphological characterization by electron microscopy of a few NPs loaded with therapeutic toxins for SGT purposes.

## 6. Extracellular Vesicles as Delivery System for SGT

Extracellular vesicles (EVs) are nanosized lipid bilayer particles produced by cells for inter-cellular communications and are present in biological fluids. They differ according to origin, biogenesis and size and contain biologically relevant cargos that impart regulatory changes in target cells [64]. Exosomes are classically considered the smallest vesicles secreted by cells, with approximate size ranging from 30 to 150 nm. They originate from intraluminal budding within multivesicular bodies (MVBs) and are released upon fusion of mature MVBs with the plasma membrane. Another major class of EVs, often referred to as microvesicles or ectosomes, contains larger vesicles (100–1000 nm), which are formed by direct outward budding from the cell membrane [65]. Among a large variety of molecules contained in the EVs, such as proteins, nucleic acids and metabolites, which are therefore more stable, being protected from degradation with respect to those freely floating in the bloodstream, we may find RNAs. Various types of RNAs have been found in EVs, coding mRNA as well as non-coding, regulatory miRNAs and lncRNAs, able to be as such transferred from recipient to target cells. DNA may also be found within EVs as single or double-stranded forms, complexed with histone proteins, from genomic or even mitochondrial and plasmid origin. DNAs can be contained inside the EVs, be attached to their outer surface or even found in both the interior and exterior. The presence of DNA in larger vesicles has been documented, while its presence in secreted exosomes still remains controversial [66]. A possible reason for cells to load DNA into EVs can reside in the maintenance of the homeostasis and getting rid of dangerous/damaged genomic DNA [67,68]. The process seems to occur through a CD63-mediated DNA shuttle or emerin-mediated nucleus instability and shedding [66]. A few studies have demonstrated the EV-mediated intercellular communication by horizontal gene transfer of single- and double-stranded DNA, showing that EV-carried DNA is functional in the receiver cells [69].

The manipulation of EV content may be accomplished by two different approaches: (i) by engineering parent cells to secrete modified EVs or (ii) by directly modifying the EV content, after their isolation [64]. For instance, the feasibility of delivering ribosome inactivating proteins via EVs was proven by encapsulating saporin by electroporation into exosomes [70]. Saporin-encapsulated EVs showed a dose-dependent cytotoxicity against epidermal carcinoma cells, being slightly more active as compared to plant saporin alone [70]. In addition, its cytotoxic effect could be further increased by adding pH-sensitive fusogenic peptides, favoring the fusion of the endosomal and EV membranes inside the cells [71,72,73].

In recent years, the potential employment of EVs for DNA delivery has been investigated, even if much less when compared to small RNAs (siRNA and miRNA). Hundreds of DNA molecules per vesicle of linear DNA have been associated with EVs via electroporation, showing that loading efficiency and capacity are dependent on size, with linear DNAs shorter than 1000 bp being more efficiently associated with EVs, as compared to larger linear or plasmid DNAs [74]. In addition, larger microvesicles (MVs) encapsulate linear and plasmid DNA much better than smaller, exosome-like EVs. A critical issue is represented by the capability to transfer the DNA to be expressed in recipient cells, since a functional gene delivery through EVs was not always observed [74,75]. In particular, microvesicles, but not exosomes, derived from cells that were transfected with plasmid DNA, induced the expression of encoded reporter proteins in recipient cells [75]. DNA transfer was achieved also in vivo by exploiting tumor cell-derived MVs carrying plasmid DNA encoding Cre recombinase, which triggered luciferase expression in transgenic Cre-lox Luc reporter mice.

An application in the field of suicide cancer gene therapy has been realized by employing MV loaded with engineered minicircle DNA that encodes prodrug-converting enzymes [76]. The minicircle episomal DNA vector consists of a circular expression cassette that lacks the prokaryotic backbone, has an improved transfection efficiency with respect to the plasmid counterpart and shows a more prolonged transgene expression. Breast cancer cells transfected by lipofection with minicircle DNA encoding a thymidine kinase (TK)/nitroreductase (NTR) fusion protein were demonstrated to produce MVs containing the plasmid. Intratumoral delivery of minicircle DNA via MV followed by prodrug administration led to the death of both targeted cells and surrounding tumor cells in a mouse model of breast cancer [76].

An alternative strategy consists of the exogenous EV loading, which exploits proteins with exposed positively charged amino acids, able to load negatively charged DNA and aid in its functional delivery [62]. The plasmid coding for a nanoluciferase reporter was associated with a super-positively charged green fluorescent protein and with EVs. The resulting complex protected the plasmid from enzymatic degradation and enabled the detection of bioluminescent signals upon expression by cells [62].

On the other hand, the loading of mRNA and even more of siRNA and miRNA into EVs seems to be much easier at least due to the smaller size of these molecules as compared to plasmid DNA. Nevertheless, RNA is less stable and once released into cell cytoplasm, can be subjected to rapid degradation without being translated into protein [75]. MVs were shown to be more prone than exosomes to load mRNAs, but even though it was delivered to recipient cells, it was shown to be rapidly degraded. In a comparative analysis, MVs from cells transiently transfected with plasmid DNA encoding a reporter gene were found containing both DNA and mRNA, but only DNA was functional following transfer to recipient cells [75].

To improve the loading efficiency of mRNA into EVs, a DNA aptamer able to recognize both the AUG region of target mRNA and the CD9 zinc finger (ZF) motif, sorting the DNA aptamer-mRNA complex into CD9-ZF engineered EVs was successfully tested [63]. Pgc1α mRNA was loaded and transferred via EV to adipose cells where it was translated prompting adipocyte browning. In the same way, the delivery of interleukin-10 mRNA showed a potent anti-inflammatory effect in a mouse model of inflammatory bowel disease [63].

So far, promising results have been reported for the EV-mediated delivery of DNA and mRNA. Further studies are needed to define analytical parameters for the efficient transfer of nucleic acids for safeguarding their functionality. In case of toxin-derived DNA, it is noteworthy to express the transgene in a controlled and targeted manner, to reduce side effects and so further broaden the potential applications of EVs as delivery vehicles of therapeutics.

Figure 3 (panels C and D, see above) shows the morphological characterization by electron microscopy of a few vesicles mediated delivery of toxins for SGT.

## 7. Conclusions and Perspectives

Following the pivotal era of the antibody-mediated toxin delivery (Immunotoxins), the use of genes coding for toxic proteins or non-human enzymes has been studied, in the last twenty years, as an innovative strategy to combat cancer. Genes of bacterial or plant origin have been intensively studied as components of different delivery formulations (Figure 4) that have evolved from simple lipidic carriers to more complex nanoparticles or extracellular vesicle vectors. The use of specific promoters to induce the selective expression of toxic genes (or by directly using modified RNAs) may reduce some of the unwanted side-effects that have previously been observed with these pivotal therapies.

The great potential of chimeric molecules able to redirect a toxin moiety such as the immunotoxins led to an intense scientific and clinical investigation in the last four decades, particularly in the United States (US). Indeed, differently from Europe, American funding agencies such as National Institute of Health (NIH) trusted that these approaches could lead to better selectivity in treating cancer patients.

Among the novel anti-cancer therapies described here, the pros and cons of GDEPT clinical approaches are extensively addressed in a comprehensive review by Karjoo et al. [3]. SGT exploiting toxin domains via non-viral vectors are promising anti-cancer tools for local delivery in an accessible tumor environment, as successfully demonstrated by the pivotal clinical trial with H9-DTA that is being reported here by Lavie et al. [30]. The toxin active domains described here act both on quiescent and proliferating cells and could be potentially interchangeable. To decrease possible immunogenicity-related issues and allow for multiple administrations in the case of tumor recurrences, combined approaches by switching toxin domains can be envisaged.

The rise in NP/EV-derived therapeutics is a great step towards clinical effective use, although investigation efforts will be needed to better understand their behaviour as safe cancer therapeutics in crossing the biological barriers, especially when administered systemically. One cost-effective approach would be using LNPs or EVs-mediated delivery of modified RNAs encoding toxin active domains after surgery in minimal residue therapies aimed at clearing potentially tumorigenic cells left-over.

Europe and our national health agencies should increase their support in R&D mirroring the NIH efforts: the scientific community has shown how impactful biotechnological approaches can become when the efforts of all the stakeholders are joined to produce a new generation of vaccines under a global threat. Europe has the strength to step into R&D with greater financial investments to better support innovation and clinical translation. A good example being “ENDOSCAPE” a collaborative biotechnology project funded by the European Commission Horizon 2020 program which is dedicated to developing clinically applicable novel gene delivery technologies. This coordinated collaborative research is expected to have a major impact on the therapeutic approaches of drug delivery for clinical applications. One of the goals is to develop non-viral based technology aimed at enhancing therapeutic efficacy in a cost-effective way and strengthening the EU competitive landscape: the ENDOSCAPE opening symposium will be held in Berlin 8–9 September 2022 (Horizon 2020 funded project No 825730) and should represent a virtuous model to follow (https://endoscape-2020.eu, accessed on 25 November 2019).

Overall, these studies are launching a new challenge beyond the immunotoxin era, hopefully, towards novel clinical investigation studies.

## Figures and Tables

**Figure 1 toxins-14-00579-f001:**
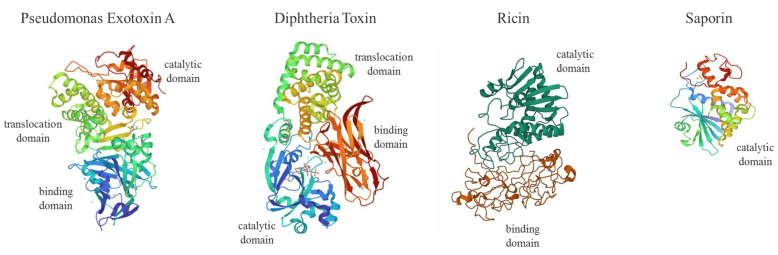
Molecular structures of Pseudomonas exotoxin A (PEA or ETA, PDB code: 1IKQ), Diphtheria Toxin (DT, 1F0L), Ricin (2AAI) and Saporin (1QI7). PEA or ETA consists in three structural domains (I) for receptor binding (blue), (II) translocation domain (green) and (III) catalytic activity (red); DT is formed by a receptor binding domain (red), translocation (green) and catalytic domain (blue) domain; type II RIP Ricin is made of two chains linked by an S-S bond displaying a galactose-binding domain (B chain; red) and A chain catalytic active domain (dark green); type I RIP Saporin contains a single N-glycosidase catalytic A domain (I). For ITX construction, the PEA binding domain I is replaced with antibody fragments. PE38 is a 38-kDa active truncated form of ETA. DT fragment A (DT-A) is a 21-kDa protein responsible for enzymatic activity. DT (Met1–Thr387) was fused to IL-2 human sequence (Ala1-Thr133) to produce the first approved ITX-derived chimera (ONTAK for T-cell lymphomas).

**Figure 2 toxins-14-00579-f002:**
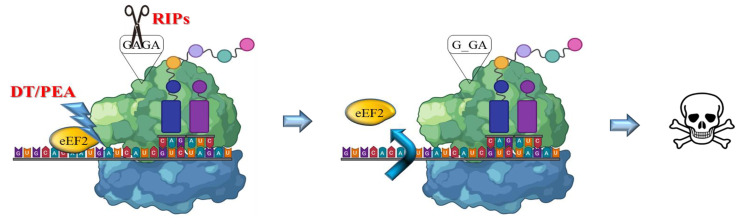
**Mechanism of action of bacterial and plant toxins.** RIPs act on 23/26/28S rRNA, by depurination of a specific adenine base in the universally conserved GAGA-tetraloop, while bacterial toxins such as DT or ETA/PEA inactivate the eukaryotic elongation factor 2 (eEF2) by ADP ribosylation using NAD^+^. Both activities trigger irreversible inhibition of protein translation, promptly leading to apoptotic cell death pathways.

**Figure 3 toxins-14-00579-f003:**
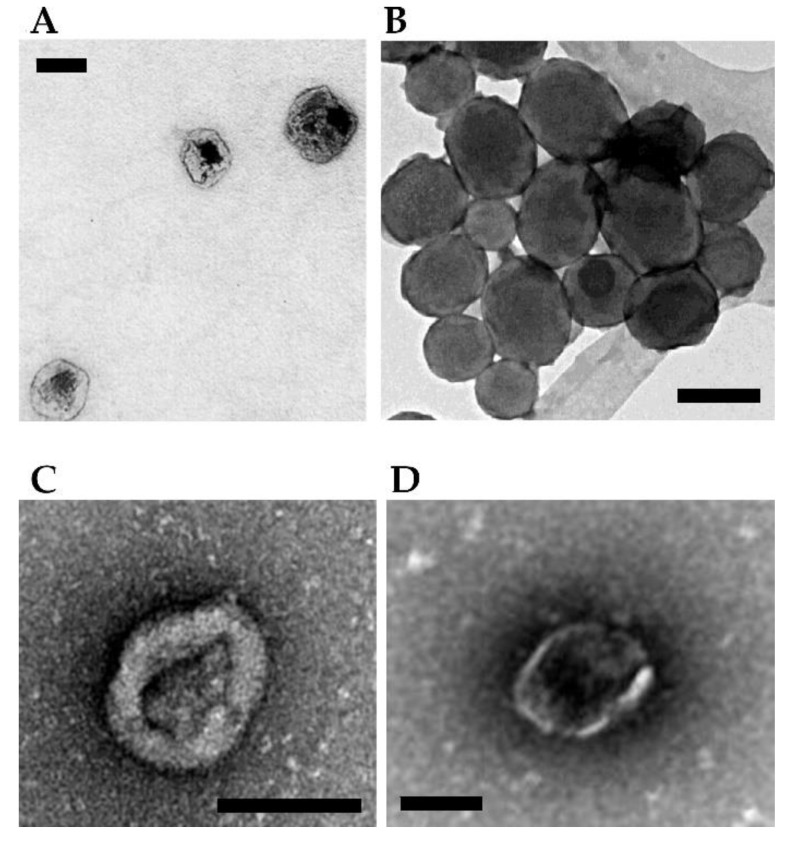
**Examples of NPs- and vesicles-based complexes for delivery of toxin genes.** (**A**) Protein-based ELR-pDNA polyplexes encoding the PAP-S toxin for treating MCF-7 breast cancer cells. Scale bar = 100 nm. Image adapted from Piña et al. [56]. (**B**) Mesoporous silica ultrasound-responsive UR-NPs@5PEI carrying the gene encoding CD and UPRT for prodrug activation against NMU cancer cells. Scale bar = 200 nm. Image adapted from Paris et al. [59]. (**C**) Minicircle DNA encoding a TK/NTR fusion protein encapsulated within MVs for treating breast cancer cells in a mouse. Scale bar = 100 nm. Image adapted from Breyne et al. [62]. (**D**) Pgc1α mRNA and interleukin-10 mRNA loaded and transferred via EV showing a potent anti-inflammatory effect in a mouse model of inflammatory bowel disease. Scale bar = 100 nm. Image adapted from Zhang et al. [63].

**Figure 4 toxins-14-00579-f004:**
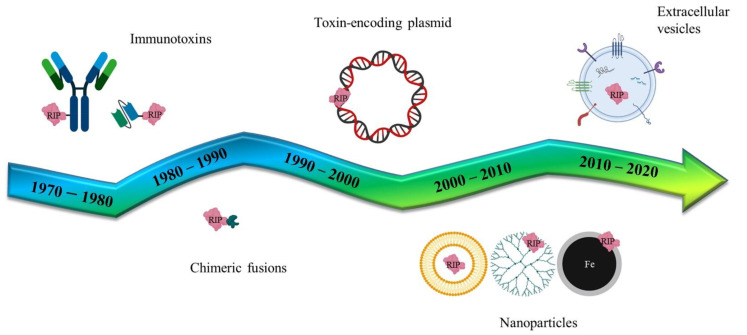
**Schematic representation of several RIP/toxin-mediated biomedical applications.** Historically, immunotoxins have been planned since the mid 1970–80s and developed for targeting the whole IgG antibody or parts of it, like the single chain fragment variants. In parallel, chimeric recombinant fusions have been studied, exploiting peptides or ligand binding-domains for tumor targeting. RIP/toxin encoding plasmids have been generated for the suicide cancer gene therapy and, more recently, artificial (nanoparticles) and natural (extracellular vesicles) versatile nanoparticulate vectors enriched the assortment of delivery systems.

**Table 1 toxins-14-00579-t001:** List of the most used enzymes in GDEPT with their sources and the enzymatic activity, bacterial and plant toxin genes used in SGT with the cited references.

Enzyme	Source	Substrate	Enzyme Activity	References
Cytosine deaminase (CD)	*Escherichia coli*	5-Fluorocytosine	Conversion of 5-Fluorocytosine to 5-Fluorouracil	[1,5]
Thymidine kinase	*Herpes simplex*	Aciclovir Ganciclovir Valganciclovir Valaciclovir	Conversiojn of Ganciclovor to Ganciclovir 3 P	[1,5]
**Bacterial Toxin**	**Source**	**Cellular Target**	**Enzyme Activity**	**References**
Clostridium perfringens enterotoxin	*Clostridium perfringens*	Claudin tight-junction protein family	Plasma membrane permeability alterations	[40]
Diphtheria toxin	*Corynebacterium diphtheria*	Elongation Factor 2	ADP-ribosyl transferase	[20,24,26,27,28,29]
Pseudomonas Exotoxin A	*Pseudomonas aeruginosa*	Elongation Factor 2	ADP-ribosyl transferase	[31,32,33,34,35,36,37]
Streptolysin O	Most strains of beta-hemolytic group *A streptococci*	Cholesterol-containing membranes	Forming rings and arcs that penetrate the apolar domain of the bilayer	[41]
**Plant Toxin**	**Source**	**Intracellular** **Target**	**Enzyme Activity**	**References**
Gelonin	*Gelonium multiflorum* L.	Ribosome	N-glycosidase	[10,50]
Abrin-A	*Abrus precatorius* L.	Ribosome	N-glycosidase	[44]
Saporin	*Saponaria officinalis* L.	Ribosome	N-glycosidase	[10,51,52,53,54,55]
Gypsophilin-S	*Gypsophila elegans* L.	Ribosome	N-glycosidase	[52]
Pokeweed Antiviral Protein	*Phytolacca americana* L.	Ribosome	N-glycosidase	[56]

## Data Availability

Not applicable.

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
