# Peer review of "From Immunotoxins to Suicide Toxin Delivery Approaches: Is There a Clinical Opportunity?"

_toxins, 2022, doi:10.3390/toxins14090579_

Round 1

Reviewer 1 Report

The paper is a bibliographic review based on 68 references as follows:

- 25% references for period 2020 – 2022

- 15% references for period 2018 – 2019

- 60% references from years 2017 and older.

This means that more than half of the references cited are older than 5 years, what is a long time, considering the novelty of the subject studied. It is understandable that there is a big body of information based on classical papers that constitute a reference for any subject, but in this case I miss more modern references.  

Other concern is that I miss some important information on targeting systems what is the Achilles’ heel of this kind of therapies. I would recommend to extend the studiy to this aspect more in detail, with more actualized bibliography.

Some of them are incomplete: example: Ref. 68 specify year

Author Response

Dear Reviewer 1,

Reviewer 2 Report

Targeted tumor therapies are more and more important in the treatment of cancer.  Protein effector molecules such as enzymes or toxins play a substantial role here.  While common strategies try to bring the protein itself into target cells, newest approaches deliver genetic information to the cell, which let the target cell produce the effector protein.  The general technique is well known due to the development of vaccines against the Corona virus.  The technique can also be applied to treat cancer in that the gene is a suicide gene, which then results in cell death when the cell is producing the toxin encoded by the gene.  In their review, the authors give an overview on bacterial and plant enzymes studied so far for their delivery and controlled expression in tumor models.

The general design of the review is convincing and thorough and important up-to-date information is provided but the review is sometimes difficult to read due to a large number of abbreviations, lack of paragraph structure, and lack of tables that summarize the information in a comparative manner.

Specific comments:

1.     Line 32–33: “a possible heterogeneous expression of the transgene should be expected in cancer cells”.  It is unclear what exactly is meant with this sentence and how the bystander effect can be mediate.  It is clear that an activated prodrug may diffuse in neighbor cancer cells that do not express the enzyme, but what does “heterogeneous expression” mean in this context?

2.     Line 52: “described by [1]” >> “described by Gholami [1]”

3.     Line 66, ref. 2–5 and line 68, ref. 6–8:  These references are very old.  This does not make them wrong but leaves the latest developments unconsidered.  At least two new references from recent time for each of the two sentences should be added.  The existing references can remain or be replaced, depending on what is suitable.

4.     Line 99–101: “immunotoxins (ITXs) [9, 10], made of a chemical or recombinant fusion between the toxin or toxic active domain and a targeting antibody or ligand binding domain.  If a ligand is used that is not an antibody or derivative thereof, then it is not an immunotoxin, but a “targeted toxin”, which is the generic term for immunotoxins, growth factor toxins, cytokine toxins and other.  The authors should clarify this.

5.     Line 119: “Denileukin diftitox or ONTAK® which was the first ITX approved by FDA (1999), recently dismissed, was followed by Tagraxofusp …”.  Neither Ontak nor Elzonris are immunotoxins, see point 4.

6.     Line 207: “In an another study …” >> “In another study …”

7.     Line 272: “… please refer to [24].” >> “… please refer to Shafiee et al. [24].”

8.     Line 394–395: “… Abrin-A isolated from Abrus precatorius (AA) have been all tested as potential “killer RNAs” in vitro and in vivo with the cmRNAs of AA …”. The abbreviation after “Abrus precatorius” is misleading. Moreover, the abbreviation is not required as used only once.  >> “… Abrin-A isolated from Abrus precatorius L. have been all tested as potential “killer RNAs” in vitro and in vivo with the cmRNAs of Abrin-A …”

9.     General: All plant names should be provided with their full botanical name, i.e., including botanic author, e.g., “Abrus precatorius” >> “Abrus precatorius L.”

10.     General: “mg kg-1” better use “mg/kg” or clarify with the editor.

11.     Chapters 3 and 4 both should include a clear table that summarizes and organizes the most important features of each targeted toxin/gene.  This is an important issue because the text, although or precisely because it contains a large amount of information, makes a direct comparison of the different approaches extremely difficult.  Such tables would be a great enrichment for the manuscript.

12.     General: Do not put a comma before “that” when “that” introduces a relative clause.

13.     General: The text contains a large number of abbreviations, which makes reading difficult. Abbreviations should be reduced, e.g., abbreviations that are never used (e.g., HRE, UMI), are only used once (e.g., MVB), or are never explained (e.g., DOTAP) should be removed.

14.     Chapter 5 is difficult to read due to missing structure.  Paragraph breaks should be added where appropriate.

15.     Including the following literature is recommended: PMID 31330822 (Yaiza Jiménez-Martínez et al.), PMID 26815223 (Maria J Piña et al.), PMID 23921576 (Houria Boulaiz).

Author Response

Dear Reviewer 2,

Reviewer 3 Report

The review summarises important parts of the current research regarding the potential of bacterial and plant toxins (or toxin subunits) in cancer treatment. It presents interesting details about different delivery approaches and their respective efficacy in various tumour models. The manuscript's general structure is relatively logical, however, subsections are partially overlapping and/or redundant, which makes reading quite difficult. For example, subsection 3.1 highlights the use of Diphtheria toxin in SGD, but lines 214ff on page 5 present research findings after the implementation of Exotoxin A in gene delivery. I would suggest significant restructuring and thoroughly revision of the text passages to generate a more precise outline that is easier to follow and makes the manuscript more concise. Further addition of more graphical elements could also be beneficial.

Author Response

Dear Reviewer 3,

Round 2

Reviewer 3 Report

The majority of suggestions from the reviewers were implemented, and the manuscript's overall structure improved. I still found some text passages hard to read. Although I wanted figures or schemes to break the long text sections, pictures of only nanoparticles are beneficial for the readers, especially with a placing at the end of the manuscript; however, the table summarising the most commonly used enzymes in SGD is clearly helpful. Concerning the "Conclusions & Perspective" section, I miss an actual conclusion, and I find the stated perspectives a bit vague. It is not clear to me how the authors would answer the question stated in the manuscript's title. 

Author Response

Dear Reviewer,

thank you for your additional suggestions. Please, note that all the new corrections have been marked in Track Change mode in the revised manuscript. According to your suggestions, we tried to improve the text passages thorughout the manuscript and moved the figure related to nanoparticles where approapriate. Furthermore, we improved the "Conclusions & Perspective" section by adding new elements to mark the current needs and opportunities about this technology.

Kind regards,

Prof. Rodolfo Ippoliti